

# Risk and prognostic nomograms for hepatocellular carcinoma with newly-diagnosed pulmonary metastasis using SEER data

Guanzhi Ye*, Lin Wang*, Zhengyang Hu, Jiaqi Liang, Yunyi Bian, Cheng Zhan and Zongwu Lin

Department of Thoracic Surgery, Zhongshan Hospital, Fudan University, Shanghai, China
* These authors contributed equally to this work.

Corresponding authors
Cheng Zhan, czhan10@fudan.edu.cn
Zongwu Lin,
lin.zongwu@zs-hospital.sh.cn

## ABSTRACT

**Purpose:** This research aimed to identify risk factors of pulmonary metastasis (PM) from hepatocellular carcinoma (HCC) and prognostic factors of patients with PM from HCC at initial diagnosis.

**Methods:** Patients diagnosed with HCC between 2010 and 2015 were reviewed retrospectively in the Surveillance, Epidemiology, and End Results (SEER) database. Patients with PM from HCC at initial diagnosis were identified from the entire cohort. Predictors for PM from HCC were identified by multivariate logistic regression analysis. Independent prognostic factors for patients with PM were determined by univariate and multivariate Cox regression analysis. Nomograms were also constructed for quantifying risk of metastasis and overall survival estimation visually.

**Results:** Our research included 30,641 patients diagnosed with HCC, of whom 1,732 cases were with PM from HCC at initial diagnosis. The risk factors causing PM from HCC were age ($P = 0.001$), race ($P < 0.001$), primary tumor size ($P < 0.001$), T stage ($P < 0.001$), N stage ($P < 0.001$), alpha-fetoprotein ($P < 0.001$), bone metastasis ($P < 0.001$), brain metastasis ($P < 0.001$), and intrahepatic metastasis ($P < 0.001$). The significantly prognostic factors for overall survival were age ($P = 0.014$), T stage ($P = 0.009$), surgical approach ($P < 0.001$), and chemotherapy ($P < 0.001$). Harrell's C-index statistics of two nomograms were 0.768 and 0.687 respectively, indicating satisfactory predictive power.

**Conclusions:** This research provided evaluation of risk factors and prognosis for patients with PM from HCC. Two nomograms we developed can be convenient individualized tools to facilitate clinical decision-making.

## INTRODUCTION

Liver cancer is one of the most aggressive malignancies and one of the major causes of cancer death globally (*Bray et al., 2018*). Hepatocellular carcinoma (HCC) accounts for 75–85% of primary liver cancer cases. Distant metastasis occurred often in patients with HCC, and lung is the most frequent location of extrahepatic metastasis comprising

approximately 30–50% of cases (*Natsuizaka et al., 2005*; *Uka et al., 2007*; *Uchino et al., 2011*; *Abbas et al., 2014*). Patients with pulmonary metastasis (PM) from HCC have an awfully unfavorable prognosis. The median overall survival (OS) was 4.5 months and 5-year OS rate was only 2.5% for synchronous HCC with PM (*Hu et al., 2018*). With the progress of therapeutic strategies for primary liver lesion and metastatic lung lesion, the survival of HCC patients with PM has been improved significantly. Therefore, it is of importance to construct metastatic risk and survival prediction assessment approaches for clinical decision-making.

Several literature reported some predictors and prognostic factors for HCC patients with PM (*Li et al., 2014*; *Huang et al., 2018*; *Lee et al., 2019*); however, it has not yet been well elucidated in population-based studies. This research aimed to identify risk factors causing PM from HCC and prognostic factors for HCC patients with PM at initial diagnosis on the basis of the Surveillance, Epidemiology, and End Results (SEER) database. Nomograms were also built as visualized tools for quantifying estimation of metastasis risk and OS.

## MATERIALS AND METHODS

### Ethics statement

This research was exempted by the Ethics Committee of Zhongshan Hospital of Fudan University (Shanghai, China), because data extracted from the publicly available SEER database were recognized as nonhuman studies (*Lu et al., 2019*).

### Patient selection

We extracted patients' data from the SEER database (http://seer.cancer.gov/), which collected cancer information from population-based cancer registries covering nearly 30% of the US population. Because information about the variable CS mets at DX-lung, which indicated the presence or absence of PM, was not available before 2010, patients diagnosed with HCC between 2010 and 2015 were finally included in our research. Exclusion criteria were as follows: (1) patients whose pathological type was not HCC; (2) patients for whom liver cancer was not their first primary tumor; (3) patients without follow-up time; (4) patients without cause of death; (5) patients with unknown race, T stage, surgical approach, bone metastasis, brain metastasis, intrahepatic metastasis, and PM. The selection process and selection codes were shown in Fig. 1 and Table S1. The old version was converted to the eighth American Joint Committee on Cancer (AJCC) TNM staging system, in which T stage reflects primary tumor size, number of primary lesions, and invasive structures of primary tumor, and N stage reflects metastasis of regional lymph nodes. The raw measurements of 30,641 patients diagnosed with HCC were provided in Data S1 and raw measurements of 1,732 HCC patients with PM were shown in Data S2.

The variables sex, age at diagnosis, race, T stage, N stage, primary tumor size, alpha-fetoprotein (AFP), surgical approach, radiation, chemotherapy, bone metastasis, brain metastasis, intrahepatic metastasis, PM, and cause-specific death classification were used in our research. The surgical approaches toward primary intrahepatic tumor included

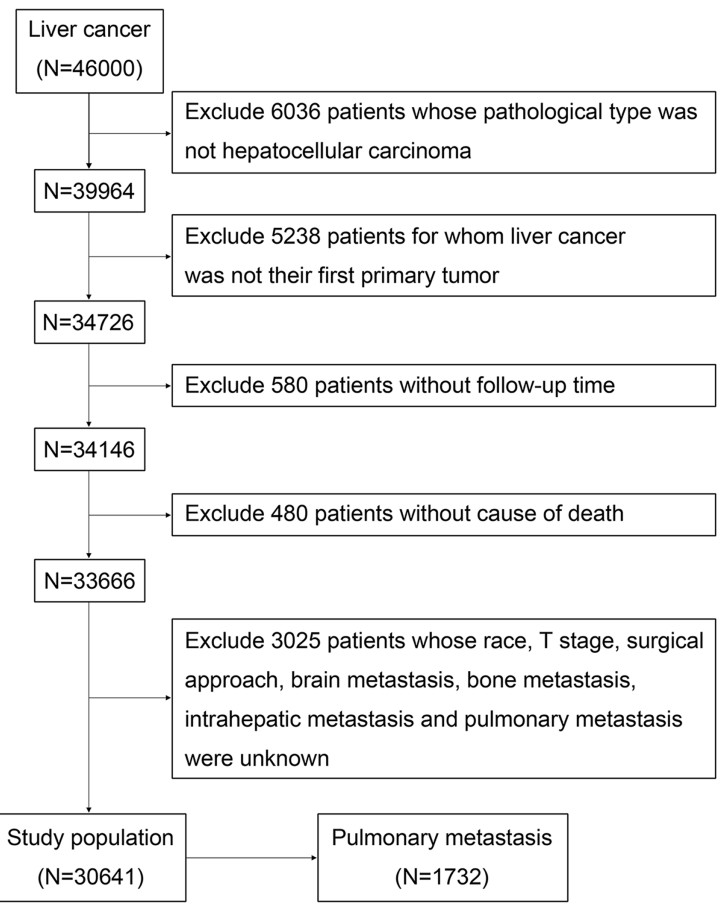

**Figure 1 The study flow diagram of the selection process.**

local treatment and liver resection, and the local treatment referred to therapeutic modalities such as transarterial chemoembolization (TACE), radiofrequency ablation, and percutaneous ethanol injection. The primary survival outcomes of the present research were OS and cancer-specific survival (CSS).

## Statistical analysis

Statistical analyses were all conducted with SPSS 21.0 statistical software (SPSS Inc, Chicago, IL, USA) and R software version 3.5.2 (*R Core Team, 2018*). The Pearson $\chi^2$ test was applied to compare categorical variables between patients with and without PM. Multivariate logistic regression analysis was adopted to determine risk factors for PM from HCC. The OS and CSS curves were compared using the Kaplan–Meier method with the Log-rank test. Univariate and multivariate analyses were conducted by the Cox regression model, and those variables with statistical significance in univariate analysis were finally included into the multivariable analysis. Nomograms based on the results of multivariate logistic regression model and Cox regression model were built, and the efficiency of prediction was estimated by Harrell's C-index statistic and calibration curves using

bootstrapping variable selection algorithms (*Austin & Tu, 2004*). All tests were two-sided, and *P*-values less than 0.05 were considered to be statistically significant.

## RESULTS

### Baseline characteristics of study population

As shown in Fig. 1, a total of 30,641 patients diagnosed with HCC were included in our research, of whom 1,732 cases (5.7%) were with PM at initial diagnosis and 28,909 cases (94.3%) were without. The median follow-up time of the entire study cohort was 8 months (interquartile range, 2–21 months). The median follow-up time for patients with and without PM were 1 month (interquartile range, 0–3 months) and 9 months (interquartile range, 2–22 months), respectively. The 1-, 2-, and 3-year OS for all patients were 50.5%, 36.2%, and 28.0%, while they were 9.3%, 4.5%, and 2.4% for patients with PM, and 53.0%, 38.1%, and 29.5% for those without PM. Table 1 showed the clinical and pathological features of the population study. Figure 2 showed the significantly shorter OS and CSS of the patients with PM from HCC, and a sudden decline of the survival curve in the first 12 months.

### Predictors for pulmonary metastasis

As shown in Table 2, the multivariate logistic regression analysis revealed that younger age ($P = 0.001$), non-white race ($P < 0.001$), higher T stage ($P < 0.001$), higher N stage ($P < 0.001$), larger primary tumor size ($P < 0.001$), elevated AFP ($P < 0.001$), and presence of bone ($P < 0.001$), brain ($P < 0.001$), and intrahepatic metastasis ($P < 0.001$) were the potential significant predictors for PM from HCC.

A nomogram for metastasis risk assessment was built on the basis of the results of multivariate logistic regression analysis above excluding surgical approach and chemotherapy which were obviously not the risk factors for PM from HCC (Fig. 3). It had a Harrell's C-index statistic of 0.768, indicating a good predictive ability for risk of PM from HCC.

### Prognostic factors for pulmonary metastasis

As shown in Table 3, univariate Cox regression analysis indicated age ($P = 0.049$), T stage ($P = 0.020$), surgical approach ($P < 0.001$), chemotherapy ($P < 0.001$), and primary tumor size ($P < 0.001$) as prognostic factors for HCC patients with PM at initial diagnosis. Multivariate Cox regression analysis based on the results above revealed that older age ($P = 0.014$), higher T stage ($P = 0.009$), absence of surgery toward primary liver lesion ($P < 0.001$), absence of chemotherapy ($P < 0.001$) were associated with a worse prognosis for HCC patients with PM.

A prognostic nomogram on the basis of the results of multivariate analysis above was constructed (Fig. 4), which internally validated by bootstrapping in 1,000 bootstrap samples. It had a Harrell's C-index statistic of 0.687 (95% CI [0.670–0.703]), again showing a good predictive efficiency for OS of HCC patients with newly diagnosed PM. As shown in Fig. S1, the calibration curves for the probability of 1-, 2-, and 3-year OS also indicated a good consistency with the actual survival.

**Table 1 Clinical and pathological features of 30,641 patients diagnosed with hepatocellular carcinoma.**

| Variable | Without pulmonary metastasis number (%) | With pulmonary metastasis number (%) | P-value |
|---|---|---|---|
| Sex | | | 0.192 |
| Male | 22,411 (77.5) | 1,366 (78.9) | |
| Female | 6,498 (22.5) | 366 (21.1) | |
| Age (year) | | | <0.001 |
| ≤50 | 2,229 (7.7) | 186 (10.7) | |
| >50 | 26,680 (92.3) | 1,546 (89.3) | |
| Race | | | <0.001 |
| White | 20,042 (69.3) | 1,086 (62.7) | |
| Black | 4,038 (14.0) | 304 (17.6) | |
| Other | 4,829 (16.7) | 342 (19.7) | |
| T stage | | | <0.001 |
| T1 | 12,645 (43.7) | 370 (21.4) | |
| T2 | 6,370 (22.0) | 155 (8.9) | |
| T3 | 3,709 (12.8) | 349 (20.2) | |
| T4 | 3,834 (13.3) | 488 (28.2) | |
| Tx | 2,351 (8.1) | 370 (21.4) | |
| N stage | | | <0.001 |
| N0 | 25,320 (87.6) | 1,101 (63.6) | |
| N1 | 1,829 (6.3) | 337 (19.5) | |
| Nx | 1,760 (6.1) | 294 (17.0) | |
| Surgery | | | <0.001 |
| No surgery | 21,628 (74.8) | 1,697 (98.0) | |
| Local treatment | 3,185 (11.0) | 14 (0.8) | |
| Liver resection | 4,096 (14.2) | 21 (1.2) | |
| Radiation | | | 0.151 |
| No | 28,585 (98.9) | 1,719 (99.2) | |
| Yes | 324 (1.1) | 13 (0.8) | |
| Chemotherapy | | | <0.001 |
| No | 16,284 (56.3) | 1,131 (65.3) | |
| Yes | 12,645 (43.7) | 601 (34.7) | |
| Tumor size (cm) | | | <0.001 |
| ≤5 | 15,327 (53.0) | 278 (16) | |
| >5 | 10,040 (34.7) | 934 (53.9) | |
| Unknown | 3,542 (12.3) | 520 (30.0) | |
| AFP | | | |
| Normal | 6,287 (21.7) | 162 (9.4) | <0.001 |
| Elevated | 17,075 (59.1) | 1,200 (69.3) | |
| Unknown | 5,547 (19.2) | 370 (21.4) | |

(Continued)

| Variable | Without pulmonary metastasis number (%) | With pulmonary metastasis number (%) | P-value |
|---|---|---|---|
| Bone metastasis | | | <0.001 |
|     No | 27,941 (96.7) | 1,442 (83.3) | |
|     Yes | 968 (3.3) | 290 (16.7) | |
| Brain metastasis | | | <0.001 |
|     No | 28,858 (99.8) | 1,693 (97.7) | |
|     Yes | 51 (0.2) | 39 (2.3) | |
| Intrahepatic metastasis | | | <0.001 |
|     No | 28,738 (99.4) | 1,567 (90.5) | |
|     Yes | 171 (0.6) | 165 (9.5) | |

**Note:**
AFP, alpha-fetoprotein.

## DISCUSSION

Hepatocellular carcinoma is a highly invasive tumor, which is prone to distant metastasis. Meanwhile, with the development of screening modalities, more and more extrahepatic metastases have been detected. Lung is the most common site of distant metastases and prognosis of HCC patients with PM was extremely poor (*Tsai et al., 1984*; *Katyal et al., 2000*; *Tung-Ping Poon, Fan & Wong, 2000*; *Hu et al., 2018*). Our research revealed that median OS of patients with PM from HCC at initial diagnosis was 1 months and 1-, 2-, and 3-year OS rate were 9.3%, 4.5%, and 2.4%, which was consistent with the results reported in the previous literature. Despite poor prognosis of HCC patients with PM, it was believed that well-control of primary liver lesion plus treatment for pulmonary lesion such as pulmonary metastasectomy could prolong survival time of these patients (*Kitano et al., 2012*; *Takahashi et al., 2016*; *Kuo et al., 2017*; *Hu et al., 2017*). Therefore, exploring predictors of PM from HCC appears to be necessary for clinical decision-making. *Guo et al. (2019)* reported that T stage, N stage, brain metastasis, and intrahepatic metastasis were risk factors for bone metastasis from HCC. *Chen et al. (2019)* pointed out that bone metastasis was significantly associated with brain metastasis from HCC. Besides, positive AFP, bilobar HCC lesions, multiplicity of HCC lesions, and primary tumor size ≥5 cm at initial diagnosis were also the independent predictors for distant metastases of HCC (*Lee et al., 2019*; *Elmoghazy et al., 2019*; *Carr & Guerra, 2016*; *Can et al., 2014*). These indicators reflected the invasiveness of primary tumor and this was an explanation that these indicators were risk factors for extrahepatic metastases from HCC. In our research, we concluded that younger age, non-white race, larger primary tumor size, higher T stage, higher N stage, elevated AFP, and presence of bone, brain and intrahepatic metastasis were significant predictors for PM from HCC.

Our research further showed that HCC patients having PM with older age, higher T stage, absence of surgery for primary intrahepatic lesion, or absence of chemotherapy significantly had an unfavorable prognosis. On the basis of the eighth edition of AJCC staging system for liver cancer, primary tumor size is an important reference criterion for

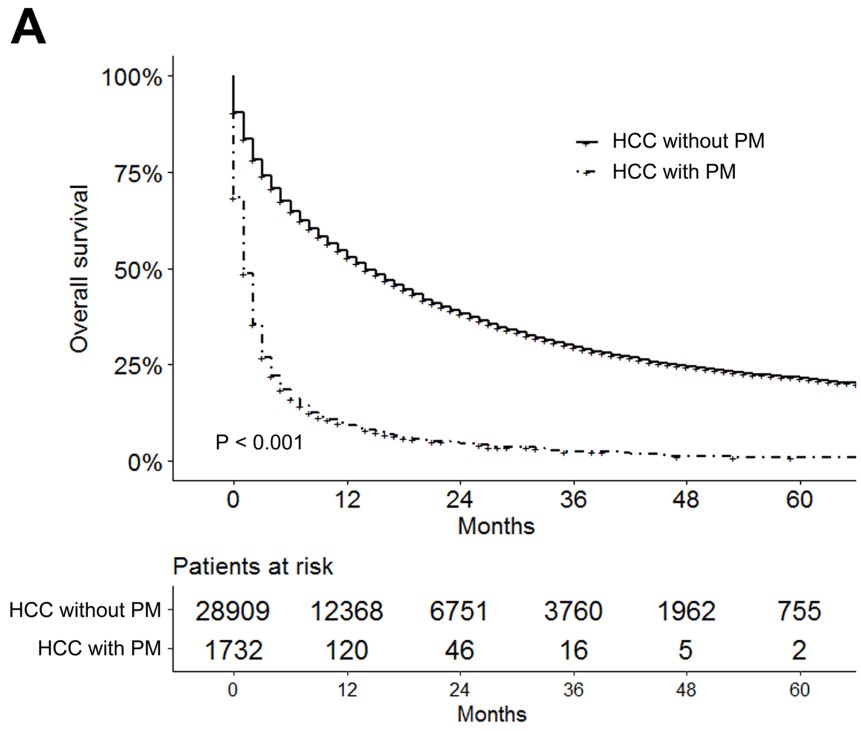

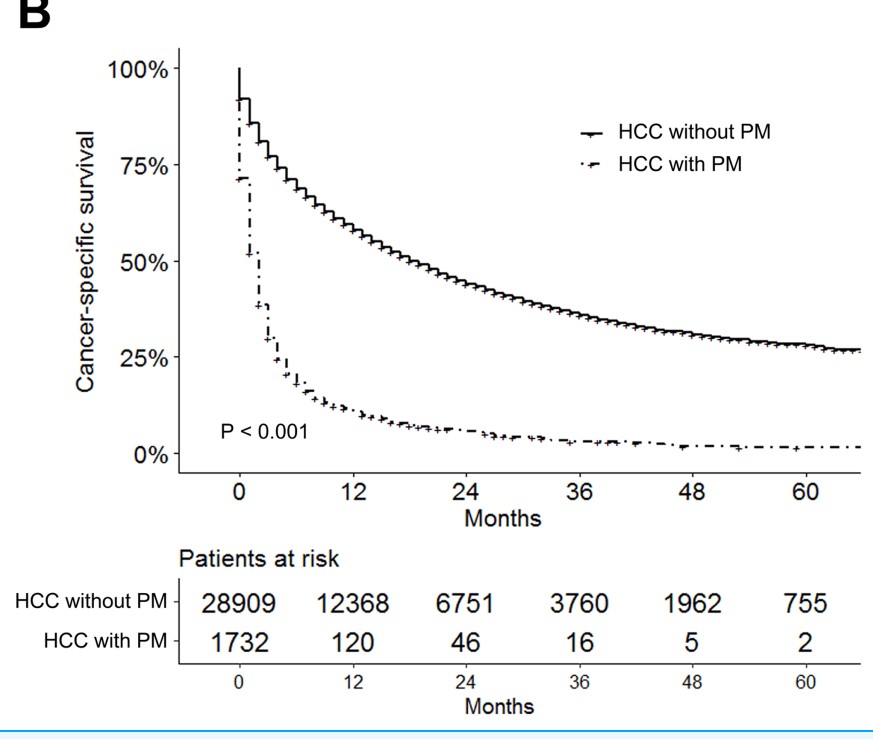

**Figure 2 Survival analysis for hepatocellular carcinoma patients with and without pulmonary metastasis.** (A) Kaplan–Meier survival curves of overall survival for hepatocellular carcinoma patients with and without pulmonary metastasis. (B) Kaplan–Meier survival curves of cancer-specific survival for hepatocellular carcinoma patients with and without pulmonary metastasis.

**Table 2 Multivariate logistic regression analysis of predictors of pulmonary metastasis from hepatocellular carcinoma.**

| Variable | OR | 95% CI | *P*-value |
|---|---|---|---|
| Age (year) | | | 0.001 |
| ≤50 | Reference | | |
| >50 | 0.756 | [0.637–0.896] | 0.001 |
| Race | | | <0.001 |
| White | Reference | | |
| Black | 1.245 | [1.083–1.433] | 0.002 |
| Other | 1.349 | [1.180–1.542] | <0.001 |
| T stage | | | <0.001 |
| T1 | Reference | | |
| T2 | 1.172 | [0.955–1.438] | 0.128 |
| T3 | 1.320 | [1.113–1.567] | 0.001 |
| T4 | 1.878 | [1.607–2.194] | <0.001 |
| Tx | 1.788 | [1.471–2.173] | <0.001 |
| N stage | | | <0.001 |
| N0 | Reference | | |
| N1 | 2.184 | [1.891–2.522] | <0.001 |
| Nx | 1.971 | [1.683–2.308] | <0.001 |
| Tumor size (cm) | | | <0.001 |
| ≤5 | Reference | | |
| >5 | 3.321 | [2.807–3.930] | <0.001 |
| Unknown | 3.722 | [3.054–4.535] | <0.001 |
| AFP | | | <0.001 |
| Normal | Reference | | |
| Elevated | 1.829 | [1.536–2.178] | <0.001 |
| Unknown | 1.535 | [1.255–1.877] | <0.001 |
| Bone metastasis | | | <0.001 |
| No | Reference | | |
| Yes | 2.976 | [2.542–3.483] | <0.001 |
| Brain metastasis | | | <0.001 |
| No | Reference | | |
| Yes | 6.906 | [4.287–11.126] | <0.001 |
| Intrahepatic metastasis | | | <0.001 |
| No | Reference | | |
| Yes | 7.743 | [6.087–9.850] | <0.001 |

Note:
OR, odd ratio; CI, confidence interval; AFP, alpha-fetoprotein.

T staging of HCC. *Huang et al. (2018)* identified intrahepatic tumor size as a major prognostic factor for HCC patients with PM undergoing liver transplantation. *Li et al. (2014)* also revealed that primary tumor size was independent prognostic factor for PM of HCC patients following hepatectomy. However, the T stage, which contains not only information about tumor size but also vascular invasion and number of primary

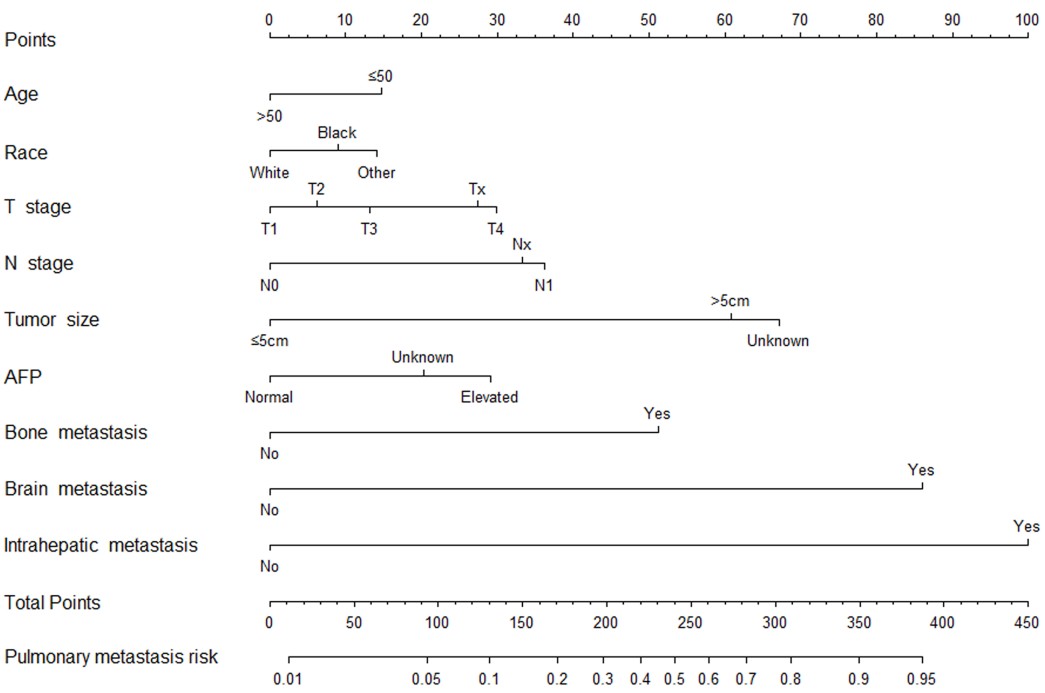

**Figure 3 Nomogram to predict pulmonary metastasis from hepatocellular carcinoma.**

tumor lesions, was more relevant to prognosis of patients than the indicator primary tumor size itself based on the AJCC staging system. *Hu et al. (2018)* showed that primary tumor size was not a prognostic factor while the T stage was in univariate analysis for OS in synchronous HCC and PM patients, although T stage did not achieve statistically significance in multivariate analysis. In our research, we found that the T stage was significantly related to OS of HCC patients with newly diagnosed PM, while primary tumor size was not. The higher T stage indicated a stronger ability of invasion and metastasis of tumor, and it might be the reason why the higher T stage led to worse prognosis.

To date, there are no standard treatment for HCC patients with PM. Because of the presence of distant metastasis, these patients lose the chance for radical surgery for primary intrahepatic tumor. However, it is widely recognized that better-control of primary HCC lesion contributes to a longer survival time. With the progress of surgical approaches, such as TACE, percutaneous ethanol injection, radiofrequency ablation, hepatectomy and liver transplantation, the control of primary HCC lesion have been improved and survival time has also been significantly prolonged for these patients. *Hu et al. (2017)* indicated that HCC patients with PM undergoing pulmonary metastasectomy treated with supportive care had a significantly worse prognosis than those received liver resection, TACE or radiofrequency ablation. *Lee et al. (2010)* reported that compared with local treatment, liver resection contributed to the prolongation of survival of HCC patients with PM undergoing pulmonary metastasectomy. *Hu et al. (2018)* revealed that HCC patients with PM at initial diagnosis undergoing liver resection survived longer than those treated with TACE or ablation.

**Table 3 Univariate and multivariate Cox regression analyses of overall survival for patients with pulmonary metastasis from hepatocellular carcinoma.**

| Variable | Univariate | | | Multivariate | | |
|---|---|---|---|---|---|---|
| | HR | 95% CI | *P*-value | HR | 95% CI | *P*-value |
| Sex | | | 0.149 | | | |
| Male | Reference | | | | | |
| Female | 0.913 | [0.807–1.033] | 0.149 | | | |
| Age (year) | | | 0.049 | | | 0.014 |
| ≤50 | Reference | | | Reference | | |
| >50 | 1.176 | [1.001–1.382] | 0.049 | 1.228 | [1.042–1.446] | 0.014 |
| Race | | | 0.987 | | | |
| White | Reference | | | | | |
| Black | 1.011 | [0.886–1.154] | 0.873 | | | |
| Other | 1.002 | [0.881–1.140] | 0.974 | | | |
| T stage | | | 0.020 | | | 0.009 |
| T1 | Reference | | | Reference | | |
| T2 | 1.039 | [0.854–1.264] | 0.703 | 1.135 | [0.916–1.407] | 0.247 |
| T3 | 1.118 | [0.958–1.305] | 0.156 | 1.222 | [1.036–1.441] | 0.017 |
| T4 | 1.230 | [1.065–1.420] | 0.005 | 1.277 | [1.105–1.477] | 0.001 |
| Tx | 1.239 | [1.062–1.446] | 0.006 | 1.043 | [0.880–1.236] | 0.628 |
| N stage | | | 0.088 | | | |
| N0 | Reference | | | | | |
| N1 | 1.141 | [1.005–1.297] | 0.042 | | | |
| Nx | 1.094 | [0.956–1.252] | 0.194 | | | |
| Tumor size (cm) | | | <0.001 | | | 0.085 |
| ≤5 | Reference | | | Reference | | |
| >5 | 1.110 | [0.964–1.279] | 0.148 | 1.104 | [0.930–1.309] | 0.258 |
| Unknown | 1.340 | [1.149–1.564] | <0.001 | 1.229 | [1.019–1.482] | 0.031 |
| AFP | | | 0.129 | | | |
| Normal | Reference | | | | | |
| Elevated | 1.191 | [1.001–1.415] | 0.048 | | | |
| Unknown | 1.199 | [0.987–1.458] | 0.068 | | | |
| Bone metastasis | | | 0.654 | | | |
| No | Reference | | | | | |
| Yes | 0.970 | [0.851–1.107] | 0.654 | | | |
| Brain metastasis | | | 0.653 | | | |
| No | Reference | | | | | |
| Yes | 1.078 | [0.778–1.494] | 0.653 | | | |
| Intrahepatic metastasis | | | 0.794 | | | |
| No | Reference | | | | | |
| Yes | 1.023 | [0.863–1.212] | 0.794 | | | |
| Surgery | | | <0.001 | | | <0.001 |
| No surgery | Reference | | | Reference | | |

| Variable | Univariate | | | Multivariate | | |
|---|---|---|---|---|---|---|
| | HR | 95% CI | *P*-value | HR | 95% CI | *P*-value |
| Local treatment | 0.386 | [0.213–0.700] | 0.002 | 0.465 | [0.255–0.847] | 0.012 |
| Liver resection | 0.292 | [0.161–0.530] | <0.001 | 0.319 | [0.175–0.580] | <0.001 |
| Radiation | | | 0.166 | | | |
| No | Reference | | | | | |
| Yes | 0.644 | [0.346–1.200] | 0.166 | | | |
| Chemotherapy | | | <0.001 | | | <0.001 |
| No | Reference | | | Reference | | |
| Yes | 0.481 | [0.432–0.536] | <0.001 | 0.478 | [0.428–0.534] | <0.001 |

**Note:**
HR, hazard ratio; CI, confidence interval; AFP, alpha-fetoprotein.

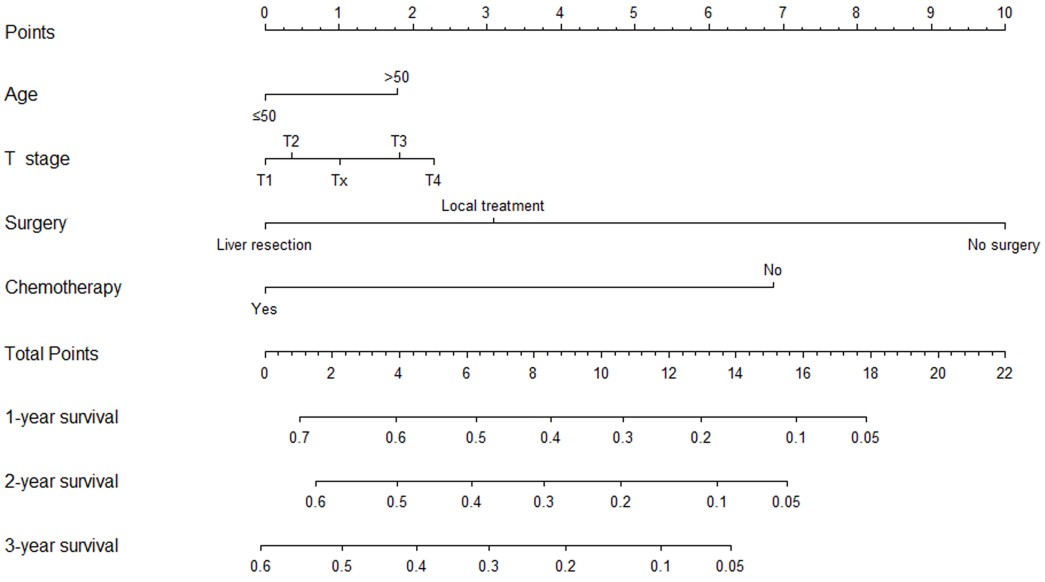

**Figure 4 Nomogram to predict 1-, 2-, and 3-year overall survival of patients with pulmonary metastasis from hepatocellular carcinoma.**

Although patient populations were different in these studies, it still demonstrated that well-control of primary HCC lesion improved survival of HCC patients with PM. Our research supported the results above, showing that surgical approach had the greatest impact on the prognosis and HCC patients with newly diagnosed PM benefited most from liver resection compared with local treatment or no surgery performed. Hence, we concluded that liver resection was a better treatment modality for HCC patients with PM who were suitable for the aggressive surgery.

There is no final conclusion as to whether patients with PM from HCC can benefit from chemotherapy, and current studies are mostly case reports with a small sample size. Several prospective studies showed that appropriate chemotherapy regimens could alleviate the patients' condition. In a clinical trial which comprised 71% of cases with PM from HCC, *Ikeda et al. (2005)* revealed that FMP chemotherapy modality (a combined treatment

of 5-fluorouracil, mitoxantrone, and cisplatin) had remarkable antineoplastic activity for patients with distant metastases. *Nakamura et al. (2008)* reported that chemotherapy of S-1 (a prodrug of 5-fluorouracil) and interferon-alpha contributed to the clinical remission of HCC with extrahepatic metastasis. *Uka et al. (2008)* also demonstrated that S-1/interferon-alpha chemotherapy modality was suitable and benefit for patients with distant metastases from HCC, especially those with PM. Due to lack of control group without chemotherapy, these studies were not able to explain the role of chemotherapy in prolonging survival of patients. However, we noted that the improvement of clinical response rate caused by chemotherapy offered patients greater opportunity to receive surgery which was believed to prolong OS. In the present research, it was obvious that patients with PM from HCC who received chemotherapy survived longer. Unfortunately, because of no information about detailed chemotherapy strategies in our data set, we were not able to compare the impact of different chemotherapy regimens on survival.

To facilitate clinical application, we established two nomograms for risk assessment of PM from HCC and survival prediction of HCC patients with PM at initial diagnosis. Both models had satisfactory predictive abilities and could be convenient individualized predictive tools for clinical decision-making. We recommended a close monitoring including detection of tumor markers and regular chest computed tomography examination for HCC patients with age less than 50 year old, non-white race, larger primary tumor size, higher T stage, higher N stage, elevated AFP, and presence of bone, brain and intrahepatic metastasis at initial diagnosis to help early detection of PM and to determine the therapeutic modalities earlier. Although there is no evidence that early detection of PM from HCC contributes to prolonging survival, we believes that early detection can be beneficial for patients as it helps management of disease and survival prediction. For HCC patients with newly diagnosed PM, we suggested aggressive therapies involving surgery toward the intrahepatic tumor or chemotherapy for highly selective patients with appropriate physical conditions and other favorable factors.

To our knowledge, this was the first population-based research focusing on the construction of metastatic risk and survival prediction approaches for patients with PM from HCC at initial diagnosis. However, limitations of the present work should be noted. Firstly, this was a retrospective study in which selection bias existed inevitably. Secondly, information about detailed treatment modalities for primary intrahepatic lesion and metastatic pulmonary lesion was not available in the SEER database and the information about pathological variable primary tumor grade was unknown in nearly 70% of the entire cohort, which was important for the multivariate analysis. Thirdly, information about performance status could not be obtained from the SEER database, which influenced therapeutic decision-making and might be a confounding factor in this study. Finally, the two nomograms we built were not validated using an external validation cohort and a further research is needed to verify our nomograms in the future.

## CONCLUSIONS

For patients with HCC, age, race, T stage, N stage, primary tumor size, AFP, and bone, brain and intrahepatic metastasis were the risk factors of PM. For patients with PM from

HCC at initial diagnosis, Age, T stage, surgical approach and chemotherapy were independent prognostic factors for OS. Nomograms we built may be individual and convenient tools for metastatic risk and prognostic assessment for PM from HCC.

### Funding

This work was supported by the National Natural Science Foundation of China (Grant No. 81572295). The funders had no role in study design, data collection and analysis, decision to publish, or preparation of the manuscript.

### Grant Disclosures

The following grant information was disclosed by the authors:
National Natural Science Foundation of China: 81572295.

### Competing Interests

Cheng Zhan is an Academic Editor for PeerJ.

### Author Contributions

- Guanzhi Ye conceived and designed the experiments, analyzed the data, contributed reagents/materials/analysis tools, prepared figures and/or tables, authored or reviewed drafts of the paper, approved the final draft.
- Lin Wang conceived and designed the experiments, analyzed the data, contributed reagents/materials/analysis tools, prepared figures and/or tables, authored or reviewed drafts of the paper, approved the final draft.
- Zhengyang Hu analyzed the data, contributed reagents/materials/analysis tools, approved the final draft.
- Jiaqi Liang contributed reagents/materials/analysis tools, prepared figures and/or tables, approved the final draft.
- Yunyi Bian contributed reagents/materials/analysis tools, prepared figures and/or tables, approved the final draft.
- Cheng Zhan conceived and designed the experiments, authored or reviewed drafts of the paper, approved the final draft.
- Zongwu Lin conceived and designed the experiments, authored or reviewed drafts of the paper, approved the final draft.

### Data Availability

The raw measurements are available in Data S1 and Data S2. The raw data shows the clinicopathological information about 30,641 patients diagnosed with hepatocellular carcinoma and 1,732 patients with pulmonary metastasis from hepatocellular carcinoma, respectively.

## Supplemental Information

Supplemental information for this article can be found online at http://dx.doi.org/10.7717/peerj.7496#supplemental-information.

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
