# Peer review of "Risk and prognostic nomograms for hepatocellular carcinoma with newly-diagnosed pulmonary metastasis using SEER data"

_PeerJ, doi:10.7717/peerj.7496_

## Round 0.1 · original submission · Minor Revisions

Overall the reviewers were satisfied with the manuscript and also raise some concerns in the comments. Please address all of the comments in point-by-point responses. Thanks

Reviewer 1 ·

Basic reporting

This study attempts to demonstrate risk factors for pulmonary metastasis (PM) from hepatocellular carcinoma (HCC) as well as poor prognosis in HCC patients with PM at initial diagnosis. The data clearly showed significant risk factors using the nomograms with large cohort. Also, their discussion seems well-written and reasonable. Since there are several points that should be revised, I would give some suggestion as following.

Experimental design

Methods seem to be well described.

Validity of the findings

1) The data seems very clear, but there is a critical point that the nomograms were not validated using external validation cohort. At this point, it is unclear whether their nomograms can work universally. Thus, they may refer this in the Discussion section.
2) Then, the data demonstrated that hepatic surgery as well as chemotherapy is a favorable prognostic factors in this cohort. However, it should be noted that performance status that could not be obtained in the SEER database can be a confounding factor that must influence therapeutic decision-making.
3) There is no evidence whether the early detection of PM can contribute to prolong survival time. So, they may mention this in the Discussion section.

Reviewer 2 ·

Basic reporting

Clear and unambiguous, professional English is used throughout.

The authors cite relevant literature, citing recent similar research.

The authors have used professional presentation structure including figures and tables.

Experimental design

The research presented here is not novel as the authors also discuss in the manuscript. However the research questions are well defined with relevant limitations of the study.

The authors should describe what T stage and N stage stand for.

Validity of the findings

The authors have assessed the impact and novelty of the study and they discuss the associated limitations.

·

Basic reporting

Title: (Risk and prognostic nomograms for hepatocellular carcinoma with newly diagnosed pulmonary metastasis)
● The aim clear.
● Title is informative and relevant.


Introduction:
• Clear, the research question clearly outlined.
• I would recommend saying: (Liver cancer is one of the most aggressive malignancies and one of the major cause of 44 cancer death globally) avoid the (Fourth) because of the controversy. Check the following : (Lung cancer is the most commonly diagnosed cancer (11.6% of the total cases) and the leading cause of cancer death (18.4% of the total cancer deaths), closely followed by female breast cancer (11.6%), prostate cancer (7.1%), and colorectal cancer (6.1%) for incidence and colorectal cancer (9.2%), stomach cancer (8.2%), and liver cancer (8.2%) for mortality).
• (Check Global cancer statistics 2018: GLOBOCAN estimates of incidence and mortality worldwide for 36 cancers in 185 countries (https://doi.org/10.3322/caac.21492)


Discussion:
● Discussed from multiple angles and placed into context without being overinterpreted. Answered the aims of the study. There is a good opportunity to repeat this study in future the data are well explained to be repeated by my opinion. You already referred to in the discussion to the Inclusion and exclusion criteria but I would urge you to write them separately and in very clear way so that would make our research clearer.



references:
• I would recommend using the numbers for the references in the discussion to avoid putting the article names between the brackets for less distraction and easier reading and following.

Experimental design

Methods:
• Please make comment about the following in the methods: The possible weakness in this study which I need the authors to comment about is using the (univariate and multivariate Cox regression analysis), as there are suggestions of violation of the proportional hazard assumption in the Cox regression model may lead to creating a false model. that does not include only time-independent predictive factors. For this reason, before the application of a simple hazard model is made, one has to check the above assumption and apply the Cox stratified regression model or time-dependent variable model if necessary.

Validity of the findings

Results:
● Tables and figures relevant and clearly presented.
● Table 1/Clinical and pathological features of 30641 patients diagnosed with hepatocellular carcinoma: There is an obvious the selection bias in that table, please try to make a comment about how we reduced that possible bias to defend and make our case stronger.

Additional comments

Overall:
● I like the idea of the research and it is new and having some creativity.
● It would be good add for what we now about the prognosis of hepatocellular carcinoma with newly diagnosed pulmonary metastasis. But they need to work on covering some bases to make the study stronger and more convincing. Would recommend to consider publication after some changes and corrections.

---

## Round 0.2 · accepted · Accept

We are happy to inform you that your manuscript is accepted for publication in PeerJ. Thank you to contribute this interesting study!

·

Basic reporting

They did a reasonable changes and for me it looks ready for publication.

Experimental design

I agree with the changes they did

Validity of the findings

Ready